Evidence synthesis  

ecology

species richness, carnivory, isotopes, rural, human footprint index, trophic

**Author for correspondence:**
Siria Gámez
e-mail: siria.gamez@yale.edu

# Downtown diet: a global meta-analysis of increased urbanization on the diets of vertebrate predators

Siria Gámez[1], Abigail Potts[2], Kirby L. Mills[2], Aurelia A. Allen[2], Allyson Holman[3], Peggy M. Randon[2], Olivia Linson[4] and Nyeema C. Harris[1]

[1]Applied Wildlife Ecology Laboratory, School of the Environment, Yale University, 195 Prospect Street, New Haven, CT 06511, USA
[2]Ecology and Evolutionary Biology, [3]School for Environment and Sustainability, and [4]College of Literature, Science and the Arts, University of Michigan, 500 S State Street #2005, Ann Arbor, MI 48109, USA

SG, 0000-0003-1496-1665; KLM, 0000-0001-7693-9629; PMR, 0000-0001-8640-7159; NCH, 0000-0001-5174-2205

Predation is a fundamental ecological process that shapes communities and drives evolutionary dynamics. As the world rapidly urbanizes, it is critical to understand how human perturbations alter predation and meat consumption across taxa. We conducted a meta-analysis to quantify the effects of urban environments on three components of trophic ecology in predators: dietary species richness, dietary evenness and stable isotopic ratios (IRs) ($\delta^{13}$C and $\delta^{15}$N IR). We evaluated whether the intensity of anthropogenic pressure, using the human footprint index (HFI), explained variation in effect sizes of dietary attributes using a meta-regression. We calculated Hedges' $g$ effect sizes from 44 studies including 11 986 samples across 40 predatory species in 39 cities globally. The direction and magnitude of effect sizes varied among predator taxa with reptilian diets exhibiting the most sensitivity to urbanization. Effect sizes revealed that predators in cities had comparable diet richness, evenness and nitrogen ratios, though carbon IRs were more enriched in cities. We found that neither the 1993 nor 2009 HFI editions explained effect size variation. Our study provides, to our knowledge, the first assessment of how urbanization has perturbed predator–prey interactions for multiple taxa at a global scale. We conclude that the functional role of predators is conserved in cities and urbanization does not inherently relax predation, despite diets broadening to include anthropogenic food sources such as sugar, wheat and corn.

## 1. Introduction

Predation is a process that underpins ecological and evolutionary dynamics at various scales, from the individual to the ecosystem. Predation can increase regional species richness and diversity by mediating competition in prey species [1,2]. Moreover, predators alter ecosystem-level processes such as nutrient cycling by provisioning carcasses and enriching soil or water columns [3]. Apart from consumptive effects, predation can structure communities indirectly through trophic cascades [4]. The fear of predation itself can engender non-consumptive effects that alter space use and aggregation of prey, subsequently driving vegetation patterns [5,6]. While the theoretical and empirical literature is rich with studies quantifying the effects of predation in natural systems, our understanding of how urban environments affect predation remains limited, even contradictory. For example, predation rates on human landscapes can be amplified by increased prey densities or relaxed because of an abundance of easily accessible anthropogenic subsidies, creating an urban predation paradox [7,8].

Cities are an emerging socio-ecological ecosystem inducing novel interactions, behavioural shifts and evolutionary trajectories [9–11]. By 2030, more

than 60% of the world's human population is projected to live in an urban area [12]. The effects on the landscape from such rapid urbanization are profound; globally, urban land cover is projected to increase by 1.2 million $km^2$ by 2030, decimating available habitat for wildlife and reducing agricultural land by 550 000 $km^2$, an area roughly the size of France [13,14]. Urbanization can decrease prey species richness and genetic diversity and modify community composition; thus, altering resource availability and diet selection for secondary and tertiary consumers [15–17]. Regional species pools are further filtered by urban form and history, novel urban species interactions and disparate distributions of natural resources in the urban landscape because of systemic racism and historical stratification of resources and environmental amenities based on race and class in urban areas [18,19]. Pollutants can concentrate in urban areas, making their way into the food web to potentially disrupt biochemical pathways in wildlife and cause disease as they do in humans [20]. Additionally, human food subsidies increase trophic niche overlap in terrestrial carnivores, potentially resulting in greater interspecific competition [21]. Carbon and nitrogen isotopic ratios (IRs) present in scat, hair and vibrissae allow researchers to assess individual heterogeneity in the trophic niche space [22,23]. Higher $\delta^{13}C$ IR reflects consumption of plants with C4 photosynthetic pathways such as corn (*Zea mays*) and sugarcane (*Saccharum* spp.), which are common in anthropogenic food sources, in contrast to lower values associated with C3 plants found in rural or wildland habitat [24]. High $\delta^{15}N$ IR indicates consumption of protein-rich animal prey, denoting trophic status and degree of carnivory [25,26]. Cities also modify wildlife behaviour, influencing vulnerability to predation or access to food, by disrupting diel patterns and vigilance behaviours [27,28].

Urbanization is a complex anthropogenic process that can alter predator–prey interactions through mechanisms associated with changes to food availability, habitat connectivity, vegetation density, and microclimate [29,30]. Anthropogenic infrastructures bisect habitat, increase the cost and mortality risk of movement and create novel temperature gradients, driving changes to population and community-level processes including predation [31–33]. Urban ecosystems are often a mosaic patchwork of suitable habitat, with both natural and managed greenspaces, which can bolster the persistence of some urban-dwelling species [34]. However, habitat fragmentation can reduce prey abundance broadly or disproportionately increase abundance for a small number of prey species, affecting predator diet selection and evenness [7,35]. In particular, roads cause a significant proportion of wildlife mortality, upwards of 49% of all adult and juvenile mortality for some species, underlying the trend of negative population growth in urban areas [36]. In North America, road-related mammal mortality increased up to 12% over the past 50 years [37]. The decline in prey species richness in cities could increase dietary overlap for interspecific competitors, resulting in competitive exclusion [21,38]. In addition to fragmentation, extensive homogenization of urban vegetation structure can reduce overall cover and affect prey behaviour and space use [39].

The effects induced from urbanization manifest differently among taxa with wildlife responses being scale-dependent [40]. For example, cities exhibit extreme temperature gradients, resulting in varied consumptive patterns, as low temperatures increased attack rate in *Daphnia* [41], while higher temperatures increased the prey consumption rate in fishes [42] and reptiles [43]. Herpetofauna are more susceptible to higher disease prevalence and pollution in urbanized ecosystems than mammalian fauna [44]. In some cases, urbanization can even hamper the spread of disease because of reduced host densities in cities compared to rural areas [45,46]. While patterns of species richness and population density vary significantly across taxa, urban birds and arthropods tend towards reduced diversity and increased abundance [47,48].

Despite the recent surge of urban ecology studies employing comparative urban versus non-urban frameworks, broad-scale predation patterns across taxa remain largely unknown. Studies often focus on a single species or city, limiting inference at a broad scale. Additionally, a lack of a standardized definition of 'urban' has made cross-city comparisons challenging, coupled with varied experimental designs, sample sizes, and bias towards readily observable study organisms. Thus, we lack a systematic understanding of how the urban environment affects predator diet, how it varies across taxa, and how these effects scale with the intensity of human impact on the landscape. Further, assumptions regarding the relaxation of predation in urban spaces remain untested across multiple taxa. Here, we conducted, to our knowledge, the first global meta-analysis of how urban environments affect three aspects of predator trophic ecology: dietary species richness (DSR, hereafter 'richness'), dietary evenness (DEV, hereafter 'evenness') and trophic niche using $\delta^{13}C$ and $\delta^{15}N$ IRs (figure 1). We used the human footprint index (HFI), a globally available metric quantifying anthropogenic pressures on the landscape [49], to explain the variability of observed effect sizes. Our aim was to inform urban ecology theory as well as guide future research, natural resource management efforts and urban planning, particularly in newly urbanizing regions. Specifically, we addressed the following questions:

(i) how does urbanization affect predator diet composition? We expected a decrease in dietary richness, given documented reductions in species richness owing to anthropogenic perturbations in urban environments [50,51]. If species richness in cities declines, the relative abundance of some common urban prey species can increase significantly. Therefore, we expected DEV to decrease in urban areas [52]. We expected the urban predator trophic niche to reflect lower $\delta^{15}N$ and greater $\delta^{13}C$ ratios relative to their rural counterparts because of a shift towards anthropogenic food sources rich in corn, wheat, and sugar [21,53,54];

(ii) how do the effect sizes of urbanization on dietary attributes differ across predator taxa? We expected trophic responses to urbanization to differ significantly among predator taxa, owing to implicit differences in diet plasticity, behaviour and natural history as well as biased representation of taxa in urban ecology literature [55,56]. Such variation in sensitivity to perturbations in urban ecosystems may ultimately drive heterogeneity in the direction and significance of effect sizes among wildlife taxa [57,58]; and

(iii) how do urban effects on predation relate to the HFI? We expected the effect sizes to positively correlate with the HFI, indicating that as the intensity of anthropogenic pressure increased, the magnitude of change

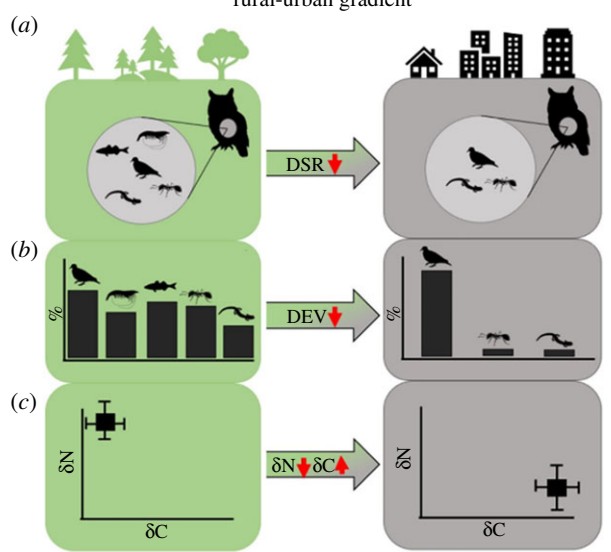

**Figure 1.** Conceptual diagram illustrating how cities can influence three components of predator trophic ecology: (a) dietary species richness (DSR), (b) dietary evenness (DEV) and (c) $\delta^{13}$C and $\delta^{15}$N isotopic ratios. Green (left) column represents rural and wildland habitat, while grey (right) denotes urban habitat. (Online version in colour.)

to predation would increase. Predator diversity and density, and thus predation rates and prey selection, are correlated with the components used to calculate the HFI, such as human population density and land-use conversion, underpinning our expectation that the degree of urbanization would explain variation in predator diet effect sizes [59,60].

# 2. Material and methods

## (a) Literature search

We completed a comprehensive literature search of empirical studies that provided estimates on aspects of predation rate, prey availability versus selection or prey diversity in the context of a predator of any taxa and compared these dietary metrics in an urban versus rural or wildland framework (discrete or gradient). Firstly, we conducted a broad topic search using the Web of Science publication database with the following terms: 'predat* AND urban', 'prey AND urban', 'prey AND urban AND rural'. We then conducted a second, stricter search by filtering results by topic 'Ecology', 'Zoology', 'Biodiversity Conservation', 'Environmental Sciences' or 'Ornithology', using the keywords: 'resource AND use AND urban', 'predat* AND urban AND wildland' and 'carniv* AND urban AND diet'. We performed a subsequent targeted search to improve representation for amphibians and reptiles. Web of Science lists publications from both United States-based and international journals, although an English language abstract or title is needed to appear in the search results. We did not limit the inclusion of studies based on year of publication or whether urban and rural samples were collected in the same year if they satisfied our other selection criteria. All vertebrate taxa were considered for inclusion. Searches were conducted iteratively (electronic supplementary material, figure S1); we recorded the number of results for each search iteration following the Preferred Reporting Items for Systematic Reviews and Meta-Analyses guidelines [61].

How studies define urban and rural varies considerably [62]. Some studies used a categorical approach, while others used a continuous gradient of urbanization based on an index such as per cent impervious surface, human population or distance to the urban core. Without a unified definition of 'urban' or use of standardized response variables in the urban ecology literature, we defined the selection criteria for inclusion in the analysis in that prospective studies needed: (i) direct measures of diet composition through observation, scat, pellets or necropsy, and (ii) summary data of diet metrics for both urban and rural categories to calculate effect sizes. In studies that sampled urban versus rural in a discrete fashion, we extracted values for each category. We extracted predation and prey composition values from the two extremes for studies that used a gradient approach. The term 'rural' is used broadly in the literature to describe non-urban habitat and implies a lack of built infrastructure. However, agricultural landscapes have some degree of anthropogenic influence via roads, buildings (e.g. barns), crop, and livestock production. To address this, we categorized the control site type as either 'wildland' or 'rural' to distinguish agricultural landscapes from less disturbed, natural habitat. We did not perform an additional subgroup analysis using control site type because of sample size limitations.

## (b) Predation consumption metrics

We explored three metrics of trophic ecology in our analyses to capitalize on the multiple methods presented in the literature. We extracted DSR data by counting each prey taxon observed in the predator's stomach contents, pellets or through direct predation observations. Species richness in a predator's diet reflects dietary breadth. We also quantified DEV by calculating the standardized niche breadth (equation 1) for urban and rural samples in each study [63]. Evenness contrasts with species richness in that the relative frequency and representation of each prey type is considered [64,65]. We recorded sample sizes and sample standard deviations where possible. However, approximately 90% of included studies had no associated measure of within-study variance. To address this limitation, we used a maximum-likelihood estimation (MLE) approach following Sangnawakij et al. [66] to estimate heterogeneity for each study's observed mean difference, enabling us to fill in missing variance values. Additional detail on estimating within-study variation when these values were not provided are described in the next section.

$$\beta = \left[ \left( \Sigma(1/\text{Prey}_{ij}^2) - 1 \right)/(n-1) \right], \quad (2.1)$$

where:
$\beta$, niche breadth; $\text{Prey}_{ij}$, fraction of items $i$ in diet that are in food category $j$ and $n$, number of possible resources.

Because studies generally depict $\delta^{13}$C and $\delta^{15}$N IR values graphically in a two-dimensional iso-scape, we extracted data of urban and rural samples from figures using the program WEB-PLOTDIGITIZER [67]. We used all three components of trophic ecology—DSR, DEV and IR—as distinct response variables in our meta-analysis to isolate predator species-level responses changes in urbanization.

## (c) Statistical analysis

Because dietary metrics were calculated post hoc for each study, and measures such as variance and sample standard deviation were not reported for all studies; we estimated within-study variation using MLE and identifiability. Briefly, this entails assuming each effect size is a random variable with a probability density function (PDF) comprised of an overall mean difference parameter ($\mu$), between-study variance ($\tau^2$) and within-study variance ($\sigma^2$) [66]. If sample sizes are not identical across studies, the PDF is considered identifiable, and the parameters of interest can be derived by taking the log-likelihood of the probability function. It is important to note that once the within-study

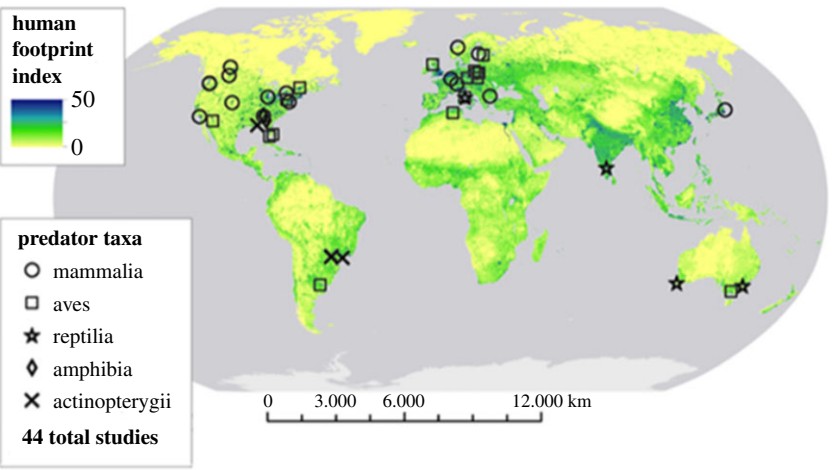

**Figure 2.** Global distribution of studies included in the analysis. Symbols represent predator taxa (class), and colour gradient illustrates human footprint index (HFI) from 2009. (Online version in colour.)

variance ($\sigma^2$) is calculated, it is assumed equal for all studies and used to calculate the effect sizes in the meta-analysis. This approach has previously been used in biomedical research; however, Sangnawakij *et al.* [66] used data simulation to demonstrate the broad applicability of this statistical technique given its efficacy in estimating within-study heterogeneity [66].

We calculated the effect size of each study using the Hedges' *g*, metric which includes a correction term for small sample sizes to determine the consequences of urbanization on predation metrics [68,69]. Positive effect sizes indicate an increase in the response variable in urban environments compared to the rural/wildland control. Between-group heterogeneity $\tau^2$ was estimated using the Sidik–Jonkman method and assumed to be equal for all predator taxa; larger $\tau^2$ values indicate greater variance of observed effect sizes between taxa [70]. Effect sizes were derived for each study, for grouped predator taxa and across all studies. Because some studies used a wildland control while others used a rural (e.g. agricultural) location to compare to the urban site, we calculated the mean *Hedges' g* richness and evenness effect size values for these two groups (mean richness$_{Rural}$: −0.19, 95% confidence interval (CI): −1.29 to 0.91; richness$_{Wildland}$: −0.16, 95% CI: −0.69 to 0.55; evenness$_{Rural}$: 0.01, 95% CI: −0.12 to 0.20; and evenness$_{Wildland}$: −0.01, 95% CI: −2.52 to 0.224). The overlapping confidence intervals indicated a lack of difference in effect sizes between the two control types and thus justified the subgroup analysis at the taxonomic level. Sample sizes for $\delta^{13}$C and $\delta^{15}$N IR values were insufficient to compare average effect size differences between control types.

We quantified the degree of anthropogenic impact by first collecting latitude and longitude coordinates from each study's urban and rural/wildland sampling sites. For studies where exact points were not reported, we extracted coordinates based on the centroid of the discrete study areas. For studies where urban-rural gradient transects were employed, we assigned coordinates at the extreme points of the study transects (e.g. urban core versus most peripheral sampling point). We then extracted HFI values for urban and rural locations for each study using ArcMap (ArcGIS Desktop v. 10.7). Urbanization is a complex process, influenced by multiple pathways of human alterations; therefore, a composite metric such as the HFI is most appropriate to capture these processes [71]. The HFI, whose values range from 0 to 50, is an effort to quantify anthropogenic pressures on the landscape by incorporating built environments, human population density, electric infrastructure, crop lands, pasture, roads, railways and navigable waterways into a single metric at a global scale [49].

We used mixed effect meta-regression models to determine how predator taxa and ΔHFI explain the variability of observed effect sizes [72]. We calculated ΔHFI as HFI$_{URBAN}$−HFI$_{RURAL}$, where positive values indicate greater anthropogenic pressures in the urban site as expected. We derived two versions of ΔHFI using the metric calculated in 1993 and later in 2009 [49]. Given the range in publication dates among papers included in our study (1986–2020), we repeated the meta-regression for both versions of ΔHFI to determine whether results were robust to when the metric was calculated. We derived a regression coefficient ($\beta$) and 95% CIs for each taxa to determine whether ΔHFI explained variation in effect sizes. Significance was determined based on whether the $\beta$ and its CI overlapped zero. To test for publication bias (e.g. asymmetry between the precision and the statistical significance of the effect sizes) in the studies included in the analysis, we built a funnel plot to visualize the overall spread of effect sizes and their corresponding error estimates (electronic supplementary material, figure S2). Finally, we performed Kendall's Rank test to quantitatively assess the correlation between effect size and error estimates. The meta-analyses were carried out in program R (v. 3.6.3) using the 'metafor' and 'meta' packages [73].

## 3. Results

### (a) Data summary

Our initial search yielded 358 studies related to predation in urban versus rural or wildland systems. Based on our selection criteria, 32 of the studies from the initial search were included in our analysis. After a subsequent search to broaden taxonomic representation, a total of 62 potential studies were found, of which nine satisfied the criteria for inclusion. Our final analyses represented a total of 44 studies with 57 effect sizes that spanned across 39 cities in six continents (figure 2; electronic supplementary material, table S1). The year of publication for included studies ranged from 1986 to 2020, with 72% ($n = 32$) of studies published after 2010. Of the 44 studies, 33 compared urban wildlife diet to that of rural (e.g. agricultural) areas while 11 a wildland (e.g. protected or wilderness area) as the reference site. One study reported diet for two species and another study reported both diet content and stable isotope ratios. In total, our meta-analysis was based on 11 986 samples of predator stomach contents, predation events, hair, and pellets across 40 species from five predatory taxa. Mammalian ($n = 19$) and avian ($n = 15$) studies

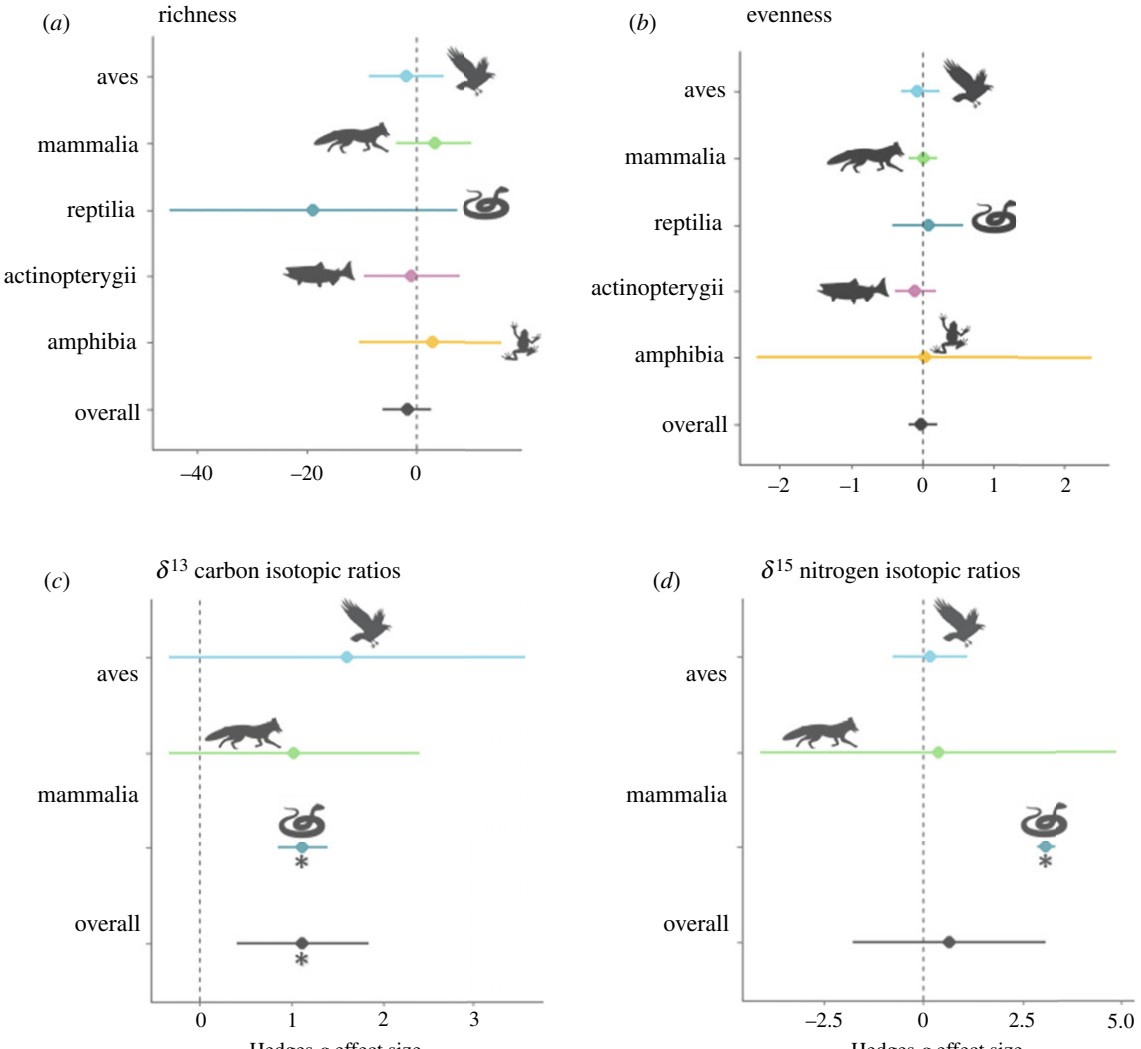

**Figure 3.** Effect sizes for components of trophic ecology grouped by taxa: (a) dietary species richness, (b) dietary evenness, (c) $\delta^{13}$C and (d) $\delta^{15}$N isotopic ratios. Hedges' g used to calculate effect sizes, and asterisk indicates significant effect of urbanization on diet metric based on whether 95% CI overlaps zero. (Online version in colour.)

had the greatest representation in our analysis, with 43.2% and 34.1%, respectively. Fishes ($n = 4$) and reptiles ($n = 4$) each comprised 9% of represented studies, while amphibians ($n = 2$) made up 4.5%. Terrestrial and aquatic systems comprised 88.6% and 11.4% of included studies, respectively.

## (b) Predation consumption metrics

We investigated consequences of urbanization on three different predation consumption metrics that represent a species' trophic ecology. Contrary to our expectations, overall DSR was not significantly lower in urban environments when studies were aggregated for all taxa (Hedges' g: −1.74, $\tau^2 = 149.08$, 95% CI: −6.08 to 2.60). The direction of the effect varied between taxa subgroups, though the effect size and respective 95% CI overlapped zero for all taxa (figure 3a). Fish, bird, and reptilian DSR decreased in response to urbanization, indicating a more specialized foraging strategy. By contrast, most mammals and amphibians consumed a greater number of prey species and adopted a more generalist foraging strategy in urban environments.

Contrary to expectations, overall DSR was also not significantly lower in cities compared to rural or wildland areas (Hedges' g: −0.02, $\tau^2 = 0.02$, 95% CI: −0.08 to 0.04). Reptiles, mammals, and amphibians showed greater DEV (i.e.

comparable representation of individual prey items) in cities (figure 3b). Conversely, bird and fish diets were skewed in urban environments, meaning a relatively small number of prey types dominated consumptive patterns. However, effect sizes and 95% CIs overlapped zero for all taxonomic subgroups.

We found evidence that diets of urban predators were more carbon-enriched, as evident by significantly a higher $\delta^{13}$C IR compared to rural predators (figure 3c,d; Hedges' g: 1.12 $\tau^2 = 0.65$, 95% CI: 0.41 to 1.84). By contrast, urban $\delta^{15}$N ratios were not significantly different than rural or wildland predators (Hedges' g: 0.67 $\tau^2 = 8.42$, 95% CI: −1.76 to 3.10). Overall, urbanization had a stronger influence on the $\delta^{13}$C ratio, signalling that predators have adopted a strategy of consuming vast quantities of anthropogenic food sources in cities, rich in sugar, corn, and wheat.

## (c) Human footprint index

Values of HFI varied greatly, even among urban sites, highlighting the breadth of intensity of anthropogenic pressures across 'urban' areas. The average HFI for urban sites was 28.7 (range = 17.3–49.4), while only 17.8 (range = 1.1–48.4) for rural sites in 1993. In comparison, anthropogenic influences estimates were higher in 2009: the average HFI for urban sites increased 1.5% (mean = 29.1; range= 18–49.4),

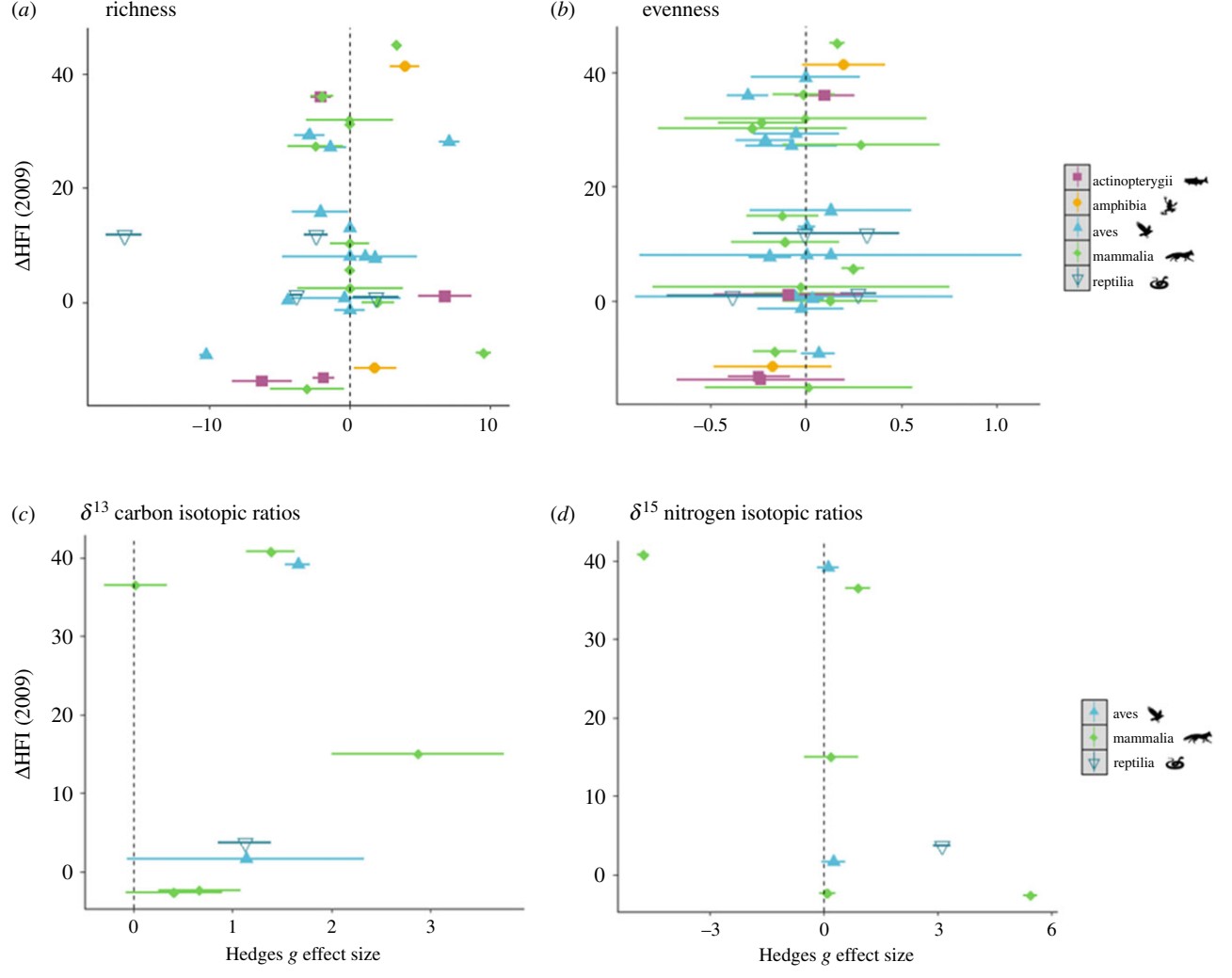

**Figure 4.** Distribution of effect sizes in response to difference in human footprint index from rural-urban sites from 2009 (ΔHFI): (a) dietary species richness, (b) dietary evenness, (c) $\delta^{13}C$ and (d) $\delta^{15}N$ isotopic ratios. (Online version in colour.)

while the average HFI for rural sites increased 13% (mean = 20.1; range = 1.3–48.4).

When effect sizes were pooled for all predator taxa, $\Delta HFI_{2009}$ did not significantly affect consumption metrics (DSR: $\beta = 0.029$, 95% CI = −0.02 to 0.08; DEV: $\beta = −0.002$, 95% CI = −0.19 to 0.15; $\delta^{13}C$: $\beta = −0.003$, 95% CI = −0.31 to 0.29; $\delta^{15}N$: $\beta = −0.002$, 95% CI = −2.1 to 2.18). Effect size responses to HFI were not distinct between studies with a rural versus a wildland control site (electronic supplementary material, figure S2). Considering how evenness varied by taxa, we found that bird evenness had a significant, negative response to $\Delta HFI_{2009}$ ($\beta = −0.004$, 95% CI = −0.008 to −0.00008). Conversely, fish DEV had a significant, positive response to $\Delta HFI_{2009}$ ($\beta = 0.007$, 95% CI = 0.003 to 0.01), though overall evenness effect size response to $\Delta HFI_{2009}$ was non-significant. Hedges' $g$ effect sizes for carbon were greater as the difference in HFI increased, however this was not the case for the other diet metrics (figure 4a–d). Therefore, we found little evidence that the degree of anthropogenic change on the landscape significantly influenced the magnitude of effect sizes of urbanization on predation.

## 4. Discussion

As urbanization alters landscapes worldwide, it is critically important to understand and anticipate the ecological consequences to wildlife living within the built environment [74,75]. Our global meta-analysis revealed that predator trophic ecology changed significantly in carbon consumption but was conserved for other diet metrics. Taxa such as amphibians face disproportionately higher extinction rates; therefore, global regions with high amphibian species richness coupled with rapid urbanization are particularly at risk of extensive changes to their faunal community composition, underscoring the importance of comparative urban-rural studies on poorly studied taxa [76].

Broad-scale shifts to DSR and DEV in predator diet owing to urbanization could have profound ecological implications for predator–prey relationships, population regulation, and disease transmission. Yet our results show that urban predators are maintaining their functional role and consuming a comparably diverse diet, even if they expanded their diet to include anthropogenic food sources. Studies used in our meta-analysis did not assess overall prey availability, which paired with predator diet could provide deeper insight into the relationship between urbanization, prey abundance, and prey selection. Further, prey population densities can in turn drive predator population increases, potentially fuelling human–wildlife conflict [52,77,78].

Urban wildlife diets are not necessarily 'protein poor' compared to rural and wildland areas, as evidenced by comparable levels of nitrogen detected in isotopic signatures.

However, consumption of carbon-based food items increased in cities, probably because of direct and indirect consumption of corn, wheat, and sugar-rich anthropogenic refuse [36] characteristic of urban habitats. Our results highlight that the ecological role of predators is not inherently relaxed in cities. Abundant prey species associated with urban environments such as brown rats (*Rattus norvegicus*), rock pigeons (*Columba livia*) and house mice (*Mus musculus*) could potentially be driving predation rates and protein (e.g. nitrogen) consumption, at least for mammalian and avian predators [52,54,79,80]. Our findings are contrary to an emerging hypothesis of relaxed predation phenomenon in urban areas, described by a meta-analysis of 25 studies that found predation rates on bird nests were reduced in urban areas [59]. Of course, relative abundance of both predators and prey in urban areas would affect predation rates and prey switching [81], and therefore experiments are necessary to fully understand the mechanisms affecting predation and meat consumption in urban and rural habitats. Additionally, consumption of meat through opportunistic pathways such as scavenging of refuse is distinct from predation, yet this process would still result in similar $\delta^{15}N$ isotope values [82]. Further experimental work is needed to disentangle these two pathways of protein enrichment.

Effect sizes did not correlate with the difference in HFI between urban and 'rural' sites, meaning the intensity of anthropogenic alterations to the landscape did not significantly amplify observed changes to predator diet. These results were robust to the year when HFI was calculated, signifying those perturbations to predator trophic ecology had probably already occurred by the time HFI was first measured in 1993 and did not change drastically after the 2009 HFI census. A possible explanation for this result is that while urban ecosystems do induce changes to predator diet, these effects occur early in the urbanization process and do not continue to amplify as cities become denser and more developed. Many cities reported low ΔHFI values, meaning the difference in their urban core and rural site was low or close to zero. Importantly, HFI values for some 'rural' sites were as high as those designated 'urban' in other studies, underscoring a fundamental challenge in urban ecology for comparative works. Moreover, such overlap in HFI between rural and urban sites demonstrates that 'rural' does not equate to 'natural' or habitats devoid of anthropogenic perturbations. Rural areas encompass agricultural production of crops and livestock, each known to alter vegetation and animal communities [83,84]. Historically, 'urban–rural' comparative studies have framed these categorizations as dichotomous spaces and carried an implicit assumption about the relatively intact character of 'rural' areas, which is not necessarily consistent at large spatial and temporal scales [62]. Further, the HFI metric is a composite indicator of human pressures and includes both the built environment and agricultural activities. Such an approach to creating a single metric to capture the effects of urbanization could contribute to low ΔHFI values in comparisons between rural and urban sites.

While our results provide key insights on predation in urban environments, we recognize limitations that require future work. We found bias in taxonomic representation with more studies on predation across urban–rural gradients for birds and mammals, particularly in North America where a small pool of species is over-represented in urban diet studies. Taxonomic bias is a well-known trend in the conservation ecology field, where charismatic vertebrate species have been historically over-represented in published studies [85]. We also found bias in the distribution of sample sizes across taxa. Molecular techniques in predation studies commonly use pellets or scat, requiring additional analysis to identify individuals in the population. Fewer than 10% of included studies identified individual host identity, potentially skewing our interpretation of the effect of urbanization on predation and limiting our inference at broad ecological scales [86]. The geographical representation of biomes in published studies was largely skewed to terrestrial ecosystems, highlighting the urgent need to study aquatic systems in proximity to urban areas. Regarding regional representation in the meta-analysis, Africa and Asia are particularly under-represented relative to their landmass and number of major cities. Such regional bias in the urban ecology literature is not indicative of a lack of research produced in these continents, but could reflect a lack of language translation tools in Western publication databases [87]. Additionally, we acknowledge that some species exhibit dietary plasticity, shifting their foraging strategies, and that this behavioural heterogeneity was not captured by our study [88]. Though we did not include considerations of omnivory versus strict carnivory in our analysis, such work would provide important insight into variation in urbanized diets. Finally, given inconsistencies in studies reporting sample variances, we highlight the need for such estimates to facilitate future cross-taxa and cross-site comparisons of urban ecology.

Wildlife must increasingly adapt to city living and our synthesis underscores how the built environment modifies a fundamental ecological process. In a rapidly urbanizing world, perturbations to predator–prey relationships can drastically change human–wildlife interactions, ecosystem processes and species extinctions; it is therefore critically important to understand these changes to inform future efforts to mitigate them. We recommend future studies aim to obtain long-term diet data at multiple sites for predatory species including at the individual level, as quantifying dietary changes over time with growing infrastructures can guide coexistence strategies for humans and wildlife [89,90]. The inclusion of stable isotopes in future urban–rural diet analyses would fill critical information gaps to couple trophic and urban ecology, as these data capture a crucial dimension of niche space at a broader trophic level. Recent work has demonstrated variation in dietary niche and trophic position for urban versus rural coyotes using stable isotopes [22]. To conclude, we provide, to our knowledge, the first quantification of predator diets in a comparative urban versus rural or wildland framework for multiple predator taxa at a global multi-city scale and reveal that urbanization enriches predator diets with carbon but does not inherently relax predation.

**Data accessibility.** The authors of this work are committed to ensuring the data used in our analysis are publicly accessible. The R code used to calculate within-study variation, forest plots and confidence intervals, as well as spreadsheets used to calculate species richness and evenness, are available from the Dryad Digital Repository: https://doi.org/10.5061/dryad.vx0k6djss [91].

**Authors' contributions.** S.G.: conceptualization, data curation, formal analysis, methodology, project administration, validation, visualization, writing—original draft, writing—review and editing; A.P.:

conceptualization, data curation, methodology, writing—original draft; K.L.M.: conceptualization, data curation, methodology, visualization, writing—original draft; A.A.A.: conceptualization, data curation, methodology, writing—original draft; A.H.: conceptualization, data curation, methodology, writing—original draft, writing—review and editing; P.M.R.: conceptualization, data curation, methodology, writing—original draft, writing—review and editing; O.L.: conceptualization, data curation, methodology, writing—original draft; N.C.H.: conceptualization, data curation, methodology, project administration, supervision, visualization, writing—original draft, writing—review and editing.

All authors gave final approval for publication and agreed to be held accountable for the work performed therein.

Competing interests. We declare we have no competing interests.

Funding. We received no funding for this study.

Acknowledgement. Our sincere thanks to members of the Applied Wildlife Ecology (AWE) Laboratory, specifically, R. Malhotra, S. Lima and N. Arringdale who assisted with data collection and provided feedback on the conceptual framework and figures in this work. We also thank Drs A. Ostling and N. Carter for their constructive comments which improved the manuscript. This meta-analysis incorporated research from across the globe on lands traditionally belonging to the Sámi, Muscogee Creek, Wajuk, Choctaw, Cree, Sioux, Skokomish, Wendake-Nionwentsïo, Calusa, Seminole, Hohokam, Tohono O'odham, Susquehannock, Wurundjeri, Mbya, Guarani, Günün, Mississauga, Ofaié, Tequesta, Taino, Ngambri, Tulalip, Tongva and Anishanaabe peoples.

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
