## [Peer Review File · Proceedings of the Royal Society B: Biological Sciences]

Review History

RSPB-2021-1205.R0 (Original submission)

Review form: Reviewer 1 (Matt W. Hayward)

Recommendation

Reject – article is scientifically unsound

Scientific importance: Is the manuscript an original and important contribution to its field?

Acceptable

General interest: Is the paper of sufficient general interest?

Acceptable

Quality of the paper: Is the overall quality of the paper suitable?

Marginal

Is the length of the paper justified?

Yes

Should the paper be seen by a specialist statistical reviewer?

No

Do you have any concerns about statistical analyses in this paper? If so, please specify them explicitly in your report.

No

It is a condition of publication that authors make their supporting data, code and materials available - either as supplementary material or hosted in an external repository. Please rate, if applicable, the supporting data on the following criteria.

Is it accessible?

Yes

Is it clear?

Yes

Is it adequate?

Yes

Do you have any ethical concerns with this paper?

No

Comments to the Author

Sorry - you've got me again. I am still not convinced that a comparison between urban areas and a merging of rural and wildland areas is a valid comparison given rural areas may be damaging the validity of the 'control' aspects of the wildland areas. As you don't provide sample sizes for each of these treatments still, I can only assume wildland sites are scant and this is predominantly a comparison between urban areas and rural areas - which doesn't really inform us about the changes urbanisation has caused on predation patterns. Your finding of no differences between urban and rural/wildlands isn't surprising given rural lands are probably filled with anthropogenically-derived food subsidies for predators. Hence, I urge you go add in sample sizes for both rural and wildland sites, and ideally compare each individually with urbanisation.

Some minor comments:

- L104-106: I think you should explain what you are comparing urban environments to.

- L231-232: How many of these 32 studies were from wildland and rural?

- L311-312: I'd add 'with rural and wildland sites' to the end of this sentence - the lack of clarity regarding the comparator is challenging in reading this paper I feel.

I hope this is helpful.

Sincerely

Matt

Review form: Reviewer 2

Recommendation

Accept as is

Scientific importance: Is the manuscript an original and important contribution to its field?

Excellent

General interest: Is the paper of sufficient general interest?

Good

Quality of the paper: Is the overall quality of the paper suitable?

Good

Is the length of the paper justified?

Yes

Should the paper be seen by a specialist statistical reviewer?

No

Do you have any concerns about statistical analyses in this paper? If so, please specify them explicitly in your report.

No

It is a condition of publication that authors make their supporting data, code and materials available - either as supplementary material or hosted in an external repository. Please rate, if applicable, the supporting data on the following criteria.

Is it accessible?

Yes

Is it clear?

Yes

Is it adequate?

Yes

Do you have any ethical concerns with this paper?

No

Comments to the Author

Dear Authors,

The manuscript was substantially improved, the Authors provided substantial explanation and addressed to comments in the previous review. I am satisfied with the corrections as well the answers to the reviewers. The manuscript could be accepted for publication.

Review form: Reviewer 3

Recommendation

Accept with minor revision (please list in comments)

Scientific importance: Is the manuscript an original and important contribution to its field?

Excellent

General interest: Is the paper of sufficient general interest?

Excellent

Quality of the paper: Is the overall quality of the paper suitable?

Excellent

Is the length of the paper justified?

Yes

Should the paper be seen by a specialist statistical reviewer?

No

Do you have any concerns about statistical analyses in this paper? If so, please specify them explicitly in your report.

No

It is a condition of publication that authors make their supporting data, code and materials available - either as supplementary material or hosted in an external repository. Please rate, if applicable, the supporting data on the following criteria.

Is it accessible?

Yes

Is it clear?

Yes

Is it adequate?

Yes

Do you have any ethical concerns with this paper?

No

Comments to the Author

The authors have done an excellent job of responding to reviewers' concerns and suggestions. The changes to Figure 4, in particular, clearly address the concerns raised by other reviewers about the lumping of "rural" and "wildland" categories together for comparison with "urban." As I had mentioned in my comments to the editor on the original submission, these concerns are important, but the solution proposed by reviewers would have required arbitrary decisions on the part of the authors. Instead, the approach the authors have taken in this revision (comparing the delta HFI to effect sizes) is much more clear and rigorous. There is also a really nice paragraph in the discussion (lines 324-343) that describes the complexity of doing comparative studies, given the complexity and heterogeneity inherent in cities' surrounding landscapes.

I have only the most minor of editorial comments on this version:

Line 304 - change "out" to "our"

Line 326 - change predator's to predators' (if I'm reading this correctly)

Line 364 - run-on sentence; needs a break between "extinctions" and "it", I think

Line 374 - change revealed to reveal (or change tense of the first part of the sentence)

Decision letter (RSPB-2021-1205.R0)

21-Jun-2021

Dear Ms Gámez:

I am writing to inform you that your manuscript RSPB-2021-1205 entitled "Downtown Diet: a global meta-analysis of urbanization on consumption patterns of vertebrate predators" has, in its current form, been rejected for publication in Proceedings B.

This action has been taken on the advice of referees, who have recommended that substantial revisions are necessary. With this in mind we would be happy to consider a resubmission, provided the comments of the referees are fully addressed. However please note that this is not a provisional acceptance.

Sincerely,
Professor Gary Carvalho
mailto: proceedingsb@royalsociety.org

Editor comments to Author:

Thank you for your resubmission of the above manuscript. You will see that we have received three responses from referees, and that two out of the three are happy with the proposed revisions made.

I continue to have two major concerns, the first relating to comments concerning the overall design and analysis of data, and the second an apparent lack of response to my specific editorial questions.

1. An original reviewer, referee# 1, continues to question a fundamental issue relating to the comparisons being made, and the validity thereof. I have also looked at the manuscript myself, and while there remains a disparity of opinions, I find the case made relating to the classification of urban information, not fully convincing. I agree, and clearly the issue raised relating to sample sizes is fundamental, and I also am unable to find these details in your manuscript. I very much appreciate the additional detail you have provided concerning the available data that you have, but there does need to be a high level of transparency, that is readily accessible, concerning the robustness of the comparisons made. It is indeed very unusual to request a second major resubmission, but I continue to see significant potential and likely impact of your work. I have classified it as a resubmission, since clearly, in terms of the inferences, and likely impact, we do need absolute confidence that despite the shortcomings, that inferences made are indeed sufficiently valid to avoid bias and any misunderstandings.

2. Below (between the margins added), I repeat part of my original editorial letter, concerning the formatting and content of evidence synthesis manuscripts, with reference to specific elements: in your response letter, and forgive me if I have somehow missed any detail, I cannot find the specific and explicit responses to the issues I raised. While of course I reviewed the manuscript more fully, I do need to know the extent to which each of these issues have been fully considered and addressed. I duplicate part of the original letter below:

 START OF EXERT FROM PREVIOUS DECISION LETTER

Crucially, as you will be aware, for an evidence synthesis manuscripts, it is vitally important that the methodology is sufficiently transparent and accessible to provide the readership with confidence in any inferences drawn. While there are of course many definitions of meta-analyses, essentially the analytical approach summarizes the results of several studies, allowing researchers and policymakers to understand both the average effect across studies and its variability, thus leading to more informed decisions about important policy issues. Currently, the methodology is not sufficiently informative, and inclusion of a clear protocol is critical given the complexity of the steps involved, and the fact that variance in even small decisions could influence the validity of the results. A coherent and informative published protocol facilitates accessibility of methods for searching and screening, the coding process and its collation, yielding a succinct preanalysis plan. I return to this further below. Please note, if you are merely undertaking a literature review to generate your evidence synthesis article, while this coincides with one of the broad 3 types published by PRSB, it does not necessarily equate to a formal meta-analysis, as indicated within your manuscript.

Much of the content of the constructive referee reports is self-evident, and I hope should you decide to resubmit your manuscript, will provide a useful guide of how to proceed. I would like to additionally draw your attention specifically to our requirements for publication of Evidence Synthesis articles. Notwithstanding, in your response to referees, I would be grateful if you would include a brief account relating to Editorial comments, on how the manuscript has been modified in relation to my brief suggestions. In particular, as you will have seen from the guidelines available for our Evidence Synthesis articles (<https://royalsocietypublishing.org/rspb/evidence-synthesis>), it is vital that the reader is able to assess the validity, robustness and objectivity of the evidence base presented. Importantly also, when putting the final touches to the article, please ensure wherever possible, that where relevant, you have addressed some of the questions below, that characterises the Evidence Synthesis article type, though I fully recognise, that many questions will only partially apply to your manuscript :

1. Is the key policy-related question(s) articulated clearly?
2. Is there a clear justification in support of policy relevance?
3. Is the likely target audience identified clearly?
4. Does the search for literature utilise a comprehensive range of sources?
5. Does the synthesis article apply clearly documented inclusion criteria to all potentially relevant studies found during the search?
6. Is a clear methodology described to avoid bias?
7. Is your study objectively weighted according to methodological quality of cited literature?
8. Are knowledge gaps and priorities clearly identified?
9. Are outcomes/recommendations tangible in terms of likely impact?
10. Are all necessary supporting information available and accessible??

Including a brief indication of how you have addressed the specific criteria above in your response letter would be most helpful. I appreciate that the volume of revision is extensive, and may go beyond what you had originally anticipated. Notwithstanding, I would hope you will find the constructive and detailed suggestions helpful in formulating a more robust and representative evidence synthesis article for resubmission. As indicated below, as in all peer review processes, the invitation to resubmit, is of course no guarantee of eventual publication, but I will do my best to exercise consistency in the remaining peer review process, by

approaching the original referees, at a minimum, though of course I am not in a position to confirm their availability.

END OF EXERT FROM PREVIOUS DECISION LETTER

I appreciate that consideration of your manuscript is taking some considerable time, and of course there is no guarantee of eventual publication. I appreciate that you cannot include data that are non-existent, but equally, I remain to be sufficiently convinced, that the design and choice of data included in your analyses, have the appropriate level of transparency and robustness. It is this latter aspect that requires your attention once more. Thank you in advance for your consideration. My plan would be to return the manuscript, based on the nature of your response to one of the original reviewers, and I will select most likely a second new reviewer, so as to provide an additional final opinion.

Reviewer(s)' Comments to Author:

Referee: 1

Comments to the Author(s)

Sorry - you've got me again. I am still not convinced that a comparison between urban areas and a merging of rural and wildland areas is a valid comparison given rural areas may be damaging the validity of the 'control' aspects of the wildland areas. As you don't provide sample sizes for each of these treatments still, I can only assume wildland sites are scant and this is predominantly a comparison between urban areas and rural areas - which doesn't really inform us about the changes urbanisation has caused on predation patterns. Your finding of no differences between urban and rural/wildlands isn't surprising given rural lands are probably filled with anthropogenically-derived food subsidies for predators. Hence, I urge you go add in sample sizes for both rural and wildland sites, and ideally compare each individually with urbanisation.

Some minor comments:

- L104-106: I think you should explain what you are comparing urban environments to.

- L231-232: How many of these 32 studies were from wildland and rural?

- L311-312: I'd add 'with rural and wildland sites' to the end of this sentence - the lack of clarity regarding the comparator is challenging in reading this paper I feel.

I hope this is helpful.

Sincerely

Matt

Referee: 2

Comments to the Author(s)

Dear Authors,

The manuscript was substantially improved, the Authors provided substantial explanation and addressed to comments in the previous review. I am satisfied with the corrections as well the answers to the reviewers. The manuscript could be accepted for publication.

Referee: 3

Comments to the Author(s)

The authors have done an excellent job of responding to reviewers' concerns and suggestions. The changes to Figure 4, in particular, clearly address the concerns raised by other reviewers about the lumping of "rural" and "wildland" categories together for comparison with "urban." As

I had mentioned in my comments to the editor on the original submission, these concerns are important, but the solution proposed by reviewers would have required arbitrary decisions on the part of the authors. Instead, the approach the authors have taken in this revision (comparing the delta HFI to effect sizes) is much more clear and rigorous. There is also a really nice paragraph in the discussion (lines 324-343) that describes the complexity of doing comparative studies, given the complexity and heterogeneity inherent in cities' surrounding landscapes.

I have only the most minor of editorial comments on this version:

Line 304 - change "out" to "our"

Line 326 - change predator's to predators' (if I'm reading this correctly)

Line 364 - run-on sentence; needs a break between "extinctions" and "it", I think

Line 374 - change revealed to reveal (or change tense of the first part of the sentence)

Author's Response to Decision Letter for (RSPB-2021-1205.R0)

See Appendix A.

RSPB-2021-2487.R0

Review form: Reviewer 2

Recommendation

Accept with minor revision (please list in comments)

Scientific importance: Is the manuscript an original and important contribution to its field?

Good

General interest: Is the paper of sufficient general interest?

Good

Quality of the paper: Is the overall quality of the paper suitable?

Good

Is the length of the paper justified?

Yes

Should the paper be seen by a specialist statistical reviewer?

No

Do you have any concerns about statistical analyses in this paper? If so, please specify them explicitly in your report.

No

It is a condition of publication that authors make their supporting data, code and materials available - either as supplementary material or hosted in an external repository. Please rate, if applicable, the supporting data on the following criteria.

Is it accessible?

Yes

Is it clear?

N/A

Is it adequate?

N/A

Do you have any ethical concerns with this paper?

No

Comments to the Author

Dear Authors,

The manuscript is interesting and brings valuable information about trophic ecology in the anthropogenic context. The Authors provided substantial explanations why they segregated rural and wildland areas and the fact that for some taxa the number of relevant results was not satisfactory.

However, I have slight suggestions and minor corrections which could be provided by the Authors. The most important is the title. "Diet" sounds too general. In my opinion the Authors should mention predation or predators, as it is the only trophic interaction studied in the work and also as the introduction begins with "predation." This implies the importance of it and the further flow of the manuscript.

L193 - 198 - I would move the majority of this description to introduction (L 106-107)

P 8 vs - use italics

P 10 L 180 - 186 - the explanation for MLE - I would included it only once in the "statistical analysis" only

L 412 - 415 - I suggest to move recommendation to the very last conclusion

I would like to see the captions for the supplementary material, there should table 1 and figures S1, S2, there is one file with graphs and one excel csv table

Review form: Reviewer 3

Recommendation

Accept with minor revision (please list in comments)

Scientific importance: Is the manuscript an original and important contribution to its field?

Excellent

General interest: Is the paper of sufficient general interest?

Excellent

Quality of the paper: Is the overall quality of the paper suitable?

Excellent

Is the length of the paper justified?

Yes

Should the paper be seen by a specialist statistical reviewer?

No

Do you have any concerns about statistical analyses in this paper? If so, please specify them explicitly in your report.

No

It is a condition of publication that authors make their supporting data, code and materials available - either as supplementary material or hosted in an external repository. Please rate, if applicable, the supporting data on the following criteria.

Is it accessible?

Yes

Is it clear?

Yes

Is it adequate?

Yes

Do you have any ethical concerns with this paper?

No

Comments to the Author

The revisions have taken an already important and impactful paper and have made it even clearer. I appreciate the thoughtful responses of the authors to the most recent set of reviews and editorial comments. I think their additions to the methods and analyses fully address the reviewers' and editor's concerns. I have only a couple of minor points of clarification that I would like to see in the new text. I emphasize again my disagreement with some of the suggestions made by reviewer #1 in particular, though I (like the authors) have become convinced that including some additional detail about the non-urban reference sites is helpful for readers.

In my previous reviews and correspondence with the editor about this study, I was critical of reviewer #1's suggestion of separating the non-urban site types into the categories of 'rural' and 'wildland.' These categories are even more poorly defined in the literature than is 'urban' (see references below). I think the authors' new analyses bear this out: i.e. the complete overlap of studies with rural and wildland sites as their non-urban reference in the effect size plots included in this new version of the manuscript. However, I think the authors have done a nice job of threading that needle - including some additional detail about how the non-urban site conditions varied among studies, while also contextualizing the inherent complexity in the discussion. Providing the summary number of studies with rural (agricultural) vs. wildland (protected areas/wilderness areas) is helpful for readers. I still maintain that the HFI, as a standardized quantitative measure of human influence, is the best way to characterize the non-urban sites and the variation among studies.

Given these new inclusions, I think it would be helpful to make two additional clarifications (just a sentence needed for each):

1. Clarify what information was used for this additional categorization of sites as rural vs. wildland. Verbal description of the sites in the original study? examination of their maps?
2. Include a summary of the HFI scores for the new rural vs. wildland categories.

References on complexity in the category "rural" in case this is helpful:

Boone, C. G., C. L. Redman, H. Blanco, D. Haase, J. Koch, S. Lwasa, H. Nagendra, S. Pauleit, S. T. A. Pickett, K. C. Seto, and M. Yokohari. 2014. Reconceptualizing land for sustainable urbanity. Padilla, B. J., and C. Sutherland. 2019. A framework for transparent quantification of urban landscape gradients. *Landscape Ecology* 34:1219–1229.

Review form: Reviewer 4

Recommendation

Major revision is needed (please make suggestions in comments)

Scientific importance: Is the manuscript an original and important contribution to its field?

Acceptable

General interest: Is the paper of sufficient general interest?

Acceptable

Quality of the paper: Is the overall quality of the paper suitable?

Acceptable

Is the length of the paper justified?

Yes

Should the paper be seen by a specialist statistical reviewer?

No

Do you have any concerns about statistical analyses in this paper? If so, please specify them explicitly in your report.

No

It is a condition of publication that authors make their supporting data, code and materials available - either as supplementary material or hosted in an external repository. Please rate, if applicable, the supporting data on the following criteria.

Is it accessible?

Yes

Is it clear?

Yes

Is it adequate?

Yes

Do you have any ethical concerns with this paper?

No

Comments to the Author

General comments

I did not review this previously, but it appears that it is a resubmission. I think there is still room for further improvement. In the present study, a global meta-analysis of the effects of urbanisation on diet patterns of vertebrate predators was conducted. The authors found difficulties in assigning 'urban' and attempted to make comparisons with 'rural' or 'natural areas'. Some recent literature shows that some urban landscapes are mosaics with patches of green spaces (natural and managed) that often allow the persistence of certain species, including predators, making comparisons more difficult. It may be better for some species to compare what was considered their natural diet with what they are feeding on in the urban landscape mosaic.

The authors' initial search yielded 358 studies covering a range of vertebrate taxa for the period (1986-2020). Concerning was how few studies were found in the southern hemisphere and that none were found for Africa. There definitely have been some urban diet studies of predators in African cities during this time. However, it is not clear if they only used studies where there was

a comparison across a gradient. Also, many of the northern hemisphere studies focus on relatively few predators but have many studies on those species, and that is not reflected in your results.

Also, one of the issues in urban environments is the increase in certain prey species (Discussion L340-343). Some species may appear more specialised in their diet (Results L277), but it is likely the availability of certain prey types (Results L284), so these cannot be assessed independently. There is no mention of increased anthropogenic pollution in the diet of some of the urban predators, but this has been reported in some studies.

I found the manuscript generally well written. Perhaps replace 'taxonomic group' with 'taxa' throughout. I suggest replacing 'due to' with 'because of' throughout. Also, replace 'compared to' with 'compared with' or 'than'. Also, where 'mammal' or 'reptile' are adjectives rather change to 'mammalian' and 'reptilian'. Generally, Figure and Table in text start with a capital. I am not sure if this Journal uses UK or US English.

Specific comments

Title: Perhaps reword to

Downtown diet: a global meta-analysis of the effects of increased urbanisation on the diets of vertebrate predators

Abstract:

You need to put the timeline that your study considered.

Consider some of the comments in general.

Introduction:

A generally good synthesis of the relevant literature. However, I think you need to mention that some of the urban landscapes are mosaics with patches of green spaces (natural and managed) that often allow the persistence of certain predatory species.

L79 I think you should mention the increase in certain prey species (Discussion L340-343) here.

L105-129 Although Journals historically used the present tense for this, there is a trend to use past tense as the work is done and it reads better.

L123 remove parentheses from HFI.

Methods:

Good explanation of methods used. My only concern was the isotope data. Did you use studies that presented these data only, or did you convert some of the other diet studies based on the typical isotopic signature? If the former, you need to present how many studies there were in your results.

L136 'Firstly'.

L148 change '&' to 'and'.

L148 You need to put the timeline that your study considered.

L160 what do you mean by 'non-zero'?

L166, 197 'analyses'.

L178 How did you account for increased anthropogenic pollution like plastic in the diet?

L187 insert Latin names.

L246 what do you mean by 'publication bias'?

Results:

A good synthesis of the data but some work is needed.

L254 over what period were 358 studies found? How many northern versus southern hemisphere studies? Why were only 32 then 44 included in your analyses?

L256 'analyses'.

L256-257 move to methods.

L259 replace 'in' with 'on'.

L259 I cannot access the Supplementary Table S1 but does that list all the species and the respective studies? Did you then group studies according to the respective vertebrate taxa? If yes, was that dominated by certain mammals or birds?

L264 Was this the number of mammalian and avian studies or the number of species of each used in the analyses?

L272 change 'hypothesis' to 'prediction'.

L278 insert 'most' before mammal'.

L277, L284 see general comments.

Discussion:

A good synthesis, but see comments under general.

L320-322 what are you actually trying to say?

References:

I have not checked if the Journal format was followed.

Decision letter (RSPB-2021-2487.R0)

05-Jan-2022

Dear Ms Gámez:

Your manuscript has now been peer reviewed and the reviews have been assessed by an Associate Editor. The reviewers' comments (not including confidential comments to the Editor) and the comments from the Associate Editor are included at the end of this email for your reference. As you will see, the reviewers and the Editors have raised some concerns with your manuscript and we would like to invite you to revise your manuscript to address them.

Research ethics:

Use of animals and field studies:

It is a condition of publication that you make available the data and research materials supporting the results in the article (<https://royalsociety.org/journals/authors/author-guidelines/#data>). Datasets should be deposited in an appropriate publicly available repository and details of the associated accession number, link or DOI to the datasets must be included in the Data Accessibility section of the article (<https://royalsociety.org/journals/ethics-policies/data-sharing-mining/>). Reference(s) to datasets should also be included in the reference list of the article with DOIs (where available).

Please submit a copy of your revised paper within three weeks. If we do not hear from you within this time your manuscript will be rejected. If you are unable to meet this deadline please let us know as soon as possible, as we may be able to grant a short extension.

Best wishes,
 Dr The Proceedings B Team
 mailto: proceedingsb@royalsociety.org

Associate Editor Board Member

Comments to Author:

Thank you for your detailed and constructive response to the previous round of editorial and referee comments. I certainly appreciate, as did the reviewers, the effort expended in the resubmitted version, as well as the helpful response to the variety of substantive issues raised. I am pleased to say that collectively, there is a consensus that the manuscript has improved significantly, and that the additional details, especially in the methodology, has generated a more rigorous and robust evidence base. I appreciate that revisions cannot extend remorselessly, though I do feel there is additional benefit to a further consideration of new comments below.

First, I acknowledge the point by point responses to my editorial comments, especially those relating to methodology and the various criteria that characterise the evidence synthesis framework. Overall, I am satisfied that you have achieved the appropriate level of information and approach. I would like to invite you to consider some remaining points, including your usage of the word "diet" in the title, and whether a more explicit link to predation or predators might be more appropriate. Your reconsideration of the use of landtype categories, esp. rural and wildland is now more substantiated, though there are suggestions, with only brief additional detail, concerning the additional categorisation of sites, and the inclusion of a summary of the HFI scores for the new rural vs. wildlife categories. In this context, you will see comments referring to the possible inclusion of mosaics with patches of green spaces, and the impact of likely lack of independence between some factors, such as diet specialisation and the availability of certain prey items. Some very constructive additional suggestions are made by the referees that I think you will find both informative and appropriate in what I envisage as a final round of revision.

It is unlikely that I will need to send your manuscript out for further review, but I would appreciate a brief uploaded response letter, with a summary of key issues and the associated changes made. Thank you again for the care and effort in generating the resubmission, your valuable aid in navigating the primary changes, and importantly, your efforts in complying with our evidence synthesis format. While you will see various suggestions made, I would hope that they would require relatively little additional time, and I hope you will be in a position to revise your manuscript in the near future. I look forward to seeing a revised version in due course.

Reviewer(s)' Comments to Author:

Referee: 4

Comments to the Author(s).

General comments

I did not review this previously, but it appears that it is a resubmission. I think there is still room for further improvement. In the present study, a global meta-analysis of the effects of urbanisation on diet patterns of vertebrate predators was conducted. The authors found difficulties in assigning 'urban' and attempted to make comparisons with 'rural' or 'natural areas. Some recent literature shows that some urban landscapes are mosaics with patches of green spaces (natural and managed) that often allow the persistence of certain species, including predators, making comparisons more difficult. It may be better for some species to compare what was considered their natural diet with what they are feeding on in the urban landscape mosaic.

The authors' initial search yielded 358 studies covering a range of vertebrate taxa for the period (1986-2020). Concerning was how few studies were found in the southern hemisphere and that none were found for Africa. There definitely have been some urban diet studies of predators in African cities during this time. However, it is not clear if they only used studies where there was a comparison across a gradient. Also, many of the northern hemisphere studies focus on

relatively few predators but have many studies on those species, and that is not reflected in your results.

Also, one of the issues in urban environments is the increase in certain prey species (Discussion L340-343). Some species may appear more specialised in their diet (Results L277), but it is likely the availability of certain prey types (Results L284), so these cannot be assessed independently. There is no mention of increased anthropogenic pollution in the diet of some of the urban predators, but this has been reported in some studies.

I found the manuscript generally well written. Perhaps replace 'taxonomic group' with 'taxa' throughout. I suggest replacing 'due to' with 'because of' throughout. Also, replace 'compared to' with 'compared with' or 'than'. Also, where 'mammal' or 'reptile' are adjectives rather change to 'mammalian' and 'reptilian'. Generally, Figure and Table in text start with a capital. I am not sure if this Journal uses UK or US English.

Specific comments

Title: Perhaps reword to

Downtown diet: a global meta-analysis of the effects of increased urbanisation on the diets of vertebrate predators

Abstract:

You need to put the timeline that your study considered.

Consider some of the comments in general.

Introduction:

A generally good synthesis of the relevant literature. However, I think you need to mention that some of the urban landscapes are mosaics with patches of green spaces (natural and managed) that often allow the persistence of certain predatory species.

L79 I think you should mention the increase in certain prey species (Discussion L340-343) here.

L105-129 Although Journals historically used the present tense for this, there is a trend to use past tense as the work is done and it reads better.

L123 remove parentheses from HFI.

Methods:

Good explanation of methods used. My only concern was the isotope data. Did you use studies that presented these data only, or did you convert some of the other diet studies based on the typical isotopic signature? If the former, you need to present how many studies there were in your results.

L136 'Firstly'.

L148 change '&' to 'and'.

L148 You need to put the timeline that your study considered.

L160 what do you mean by 'non-zero'?

L166, 197 'analyses'.

L178 How did you account for increased anthropogenic pollution like plastic in the diet?

L187 insert Latin names.

L246 what do you mean by 'publication bias'?

Results:

A good synthesis of the data but some work is needed.

L254 over what period were 358 studies found? How many northern versus southern hemisphere studies? Why were only 32 then 44 included in your analyses?

L256 'analyses'.

L256-257 move to methods.

L259 replace 'in' with 'on'.

L259 I cannot access the Supplementary Table S1 but does that list all the species and the respective studies? Did you then group studies according to the respective vertebrate taxa? If yes, was that dominated by certain mammals or birds?

L264 Was this the number of mammalian and avian studies or the number of species of each used in the analyses?

L272 change 'hypothesis' to 'prediction'.

L278 insert 'most' before mammal'.

L277, L284 see general comments.

Discussion:

A good synthesis, but see comments under general.

L320-322 what are you actually trying to say?

References:

I have not checked if the Journal format was followed.

Referee: 2

Comments to the Author(s).

Dear Authors,

The manuscript is interesting and brings valuable information about trophic ecology in the anthropogenic context. The Authors provided substantial explanations why they segregated rural and wildland areas and the fact that for some taxa the number of relevant results was not satisfactory.

However, I have slight suggestions and minor corrections which could be provided by the Authors. The most important is the title. "Diet" sounds too general. In my opinion the Authors should mention predation or predators, as it is the only trophic interaction studied in the work and also as the introduction begins with "predation." This implies the importance of it and the further flow of the manuscript.

L193 - 198 - I would move the majority of this description to introduction (L 106-107)

P 8 vs - use italics

P 10 L 180 - 186 - the explanation for MLE - I would included it only once in the "statistical analysis" only

L 412 - 415 - I suggest to move recommendation to the very last conclusion

I would like to see the captions for the supplementary material, there should table 1 and figures S1, S2, there is one file with graphs and one excel csv table

Referee: 3

Comments to the Author(s).

The revisions have taken an already important and impactful paper and have made it even clearer. I appreciate the thoughtful responses of the authors to the most recent set of reviews and editorial comments. I think their additions to the methods and analyses fully address the reviewers' and editor's concerns. I have only a couple of minor points of clarification that I would like to see in the new text. I emphasize again my disagreement with some of the suggestions made by reviewer #1 in particular, though I (like the authors) have become convinced that including some additional detail about the non-urban reference sites is helpful for readers.

In my previous reviews and correspondence with the editor about this study, I was critical of reviewer #1's suggestion of separating the non-urban site types into the categories of 'rural' and 'wildland.' These categories are even more poorly defined in the literature than is 'urban' (see references below). I think the authors' new analyses bear this out: i.e. the complete overlap of studies with rural and wildland sites as their non-urban reference in the effect size plots included in this new version of the manuscript. However, I think the authors have done a nice job of threading that needle - including some additional detail about how the non-urban site conditions varied among studies, while also contextualizing the inherent complexity in the discussion. Providing the summary number of studies with rural (agricultural) vs. wildland (protected areas/wilderness areas) is helpful for readers. I still maintain that the HFI, as a standardized

quantitative measure of human influence, is the best way to characterize the non-urban sites and the variation among studies.

Given these new inclusions, I think it would be helpful to make two additional clarifications (just a sentence needed for each):

1. Clarify what information was used for this additional categorization of sites as rural vs. wildland. Verbal description of the sites in the original study? examination of their maps?
2. Include a summary of the HFI scores for the new rural vs. wildland categories.

References on complexity in the category "rural" in case this is helpful:

Boone, C. G., C. L. Redman, H. Blanco, D. Haase, J. Koch, S. Lwasa, H. Nagendra, S. Pauleit, S. T. A. Pickett, K. C. Seto, and M. Yokohari. 2014. Reconceptualizing land for sustainable urbanity. Padilla, B. J., and C. Sutherland. 2019. A framework for transparent quantification of urban landscape gradients. *Landscape Ecology* 34:1219–1229.

Author's Response to Decision Letter for (RSPB-2021-2487.R0)

See Appendix B.

RSPB-2021-2487.R1

Review form: Reviewer 2

Recommendation

Accept as is

Scientific importance: Is the manuscript an original and important contribution to its field?

Acceptable

General interest: Is the paper of sufficient general interest?

Good

Quality of the paper: Is the overall quality of the paper suitable?

Good

Is the length of the paper justified?

Yes

Should the paper be seen by a specialist statistical reviewer?

No

Do you have any concerns about statistical analyses in this paper? If so, please specify them explicitly in your report.

No

It is a condition of publication that authors make their supporting data, code and materials available - either as supplementary material or hosted in an external repository. Please rate, if applicable, the supporting data on the following criteria.

Is it accessible?

Yes

Is it clear?

Yes

Is it adequate?

Yes

Do you have any ethical concerns with this paper?

No

Comments to the Author

Thanks for addressing my comments.

Decision letter (RSPB-2021-2487.R1)

28-Jan-2022

Dear Ms Gámez

I am pleased to inform you that your manuscript entitled "Downtown Diet: a global meta-analysis of increased urbanization on the diets of vertebrate predators" has been accepted for publication in Proceedings B.

Data Accessibility section

Open Access

Paper charges

Sincerely,

Professor Gary Carvalho

Appendix A

Yale SCHOOL OF THE ENVIRONMENT

195 Prospect Street, New Haven, CT 06511
(203) 432-5100

October [xx], 2021

Dr. Gary Carvalho

Proceedings of the Royal Society B

Dear Professor Carvalho and reviewers,

We would once again like to thank you for the thoughtful critiques and comments we received on our work “Downtown Diet: a global meta-analysis of urbanization on consumption patterns of vertebrate predators” (Previous submission numbers: RSPB-2021-1205, RSPB-2021-0390). We address the concerns outlined in the decision letter dated June 21, 2021 below. Since our last submission, we have expanded upon the methods section to better clarify the protocol to estimate within-study variance. Additional detail on inclusion criteria, consideration of whether the studies used rural or wildland sites as comparisons (as well as sample sizes) have been added. Specifically, we acknowledge from previous reviews that reviewers #2 and #3 were largely satisfied with our revision and only had minor editorial comments. Major concerns by reviewer #1 are addressed by elaborating on sample sizes and providing additional justifications for comparison groups. But firstly, we address rationale for requesting consideration for the article type “evidence synthesis” for our manuscript.

Regarding the guidelines for evidence synthesis submissions suggested in the editor’s comments:

3. Is the likely target audience identified clearly?

Gámez et al: we highlight the aim of our work as one that informs both the urban ecology academic sphere as well as natural resource managers and urban planners in rapidly developing regions [lines 106-108].

4. Does the search for literature utilize a comprehensive range of sources?

Gámez et al: we used the Web of Science publication database which draws from worldwide peer-reviewed journals in a broad range of topics [lines 138-148].

5. Does the synthesis article apply clearly documented inclusion criteria to all potentially relevant studies found during the search?

Gámez et al: we have added clarity to our previous description of the inclusion criteria for the meta-analysis including whether the candidate studies sampled from the urban and rural sites simultaneously [lines 138-148], as well as consideration of whether the study used an agricultural or actual wildland/wilderness area as the comparison to urban [lines 161-167].

6. Is a clear methodology described to avoid bias?

Gámez et al: further description on the estimation of within-study sample sites has been added to improve the transparency of our analysis protocol [203-214].

7. Is your study objectively weighted according to methodological quality of cited literature?

Gámez et al: We did not weigh studies by effect size according to their study design. Instead, all studies were included and weighted equally once we identified whether the candidate study satisfied the inclusion criteria (e.g., direct dietary measurement with identification of prey items, comparative urban vs. rural/wildland study design).

8. Are knowledge gaps and priorities clearly identified?

Gámez et al: We address how our study fills critical knowledge gaps in urban ecology throughout the introduction [e.g., lines 50-51, 93-101].

10. Are all necessary supporting information available and accessible?

Gámez et al: All supporting information is included in our submission.

Reviewer #1 comment responses:

Sorry - you've got me again. I am still not convinced that a comparison between urban areas and a merging of rural and wildland areas is a valid comparison given rural areas may be damaging the validity of the 'control' aspects of the wildland areas. As you don't provide sample sizes for each of these treatments still, I can only assume wildland sites are scant and this is predominantly a comparison between urban areas and rural areas - which doesn't really inform us about the changes urbanization has caused on predation patterns. Your finding of no differences between urban and rural/wildlands isn't surprising given rural lands are probably filled with anthropogenically-derived food subsidies for predators. Hence, I urge you go add in sample sizes for both rural and wildland sites, and ideally compare each individually with urbanization.

Gámez et al: We understand these concerns and now acknowledge this potential weakness emergent from aggregating rural and wildland in our discussion (add line numbers). We have also now clarified sample sizes for each treatment (Lines 260-261). We calculated the overall Hedges' g richness and evenness effect sizes for these two groups and determined they were not significantly different (Mean Richness_{Rural}: -0.19, 95% CI: -1.29, 0.91, Richness_{Wildland}: -0.16, [-0.69, 0.55], Evenness_{Rural}: 0.01, [-0.12, 0.20], Evenness_{Wildland}: -0.01, [-2.52, 0.224]). The overlapping confidence intervals indicated a lack of difference in effect sizes between the two control types and thus justified the subgroup analysis at the taxonomic level. Sample sizes for $\delta^{13}\text{C}$ and $\delta^{15}\text{N}$ isotopic ratio values were insufficient to compare average effect size differences between control types. In summary, of the 44 studies, 33 compared urban to rural and 11 compared urban to wildland. We would like to emphasize the limitations of our approach given that we have already subset the publications included in the analysis based on taxonomic group. The consideration of taxa is the primary objective of this work, particularly in a comparative, meta-analysis framework. Secondarily, we aimed to determine if the human footprint index could elucidate mechanisms responsible in explaining effect sizes. However, any further sub-setting of the effect sizes based on whether the comparison site is explicitly urban or rural

is not possible given our sample size of publications. Instead, to address the concerns raised, we offer the following response below.

Here we have plotted effect sizes (e.g., the mean difference in diet between urban site and the comparison site) of the various dietary components against human footprint index (HFI) and show that wildland and rural sites do not cluster separately as we would expect if rural and wildland were drastically different, as suggested by referee #1. We argue that for the purposes of comparing these sites to fully built, urban environments, grouping rural and wildland sites is a flawed but reasonable choice given the available data and still provide us important insight into impacts on predator diets. In addition, we will include these scatterplots in the supplementary material.

L104-106: I think you should explain what you are comparing urban environments to.

Gómez et al: We have added more detailed consideration of this issue in the methods [lines 160-167].

- L231-232: How many of these 32 studies were from wildland and rural?

Gámez et al: Of the 44 studies included in our analysis, 11 were wildland comparison sites while 33 were in rural [lines 260-261]. Additionally, we removed the word “final” from line 255 to avoid confusion about the total number of studies in our meta-analysis.

- L311-312: I'd add 'with rural and wildland sites' to the end of this sentence - the lack of clarity regarding the comparator is challenging in reading this paper I feel.

Gámez et al: added “compared to rural and wildland areas” [line 340].

Reviewer #2 comment responses:

Dear Authors,

The manuscript was substantially improved, the Authors provided substantial explanation and addressed to comments in the previous review. I am satisfied with the corrections as well the answers to the reviewers. The manuscript could be accepted for publication.

Gámez et al: We thank the reviewer for their comments and consideration of the changes made to our manuscript.

Reviewer #3 comment responses:

Line 304 - change "out" to "our"

Gámez et al: done [line 336].

Line 326 - change predator's to predators' (if I'm reading this correctly)

Gámez et al: done [line 356].

Line 364 - run-on sentence; needs a break between "extinctions" and "it", I think

Gámez et al: done [line 389, 390].

Line 374 - change revealed to reveal (or change tense of the first part of the sentence)

Gámez et al: done [line 408].

Appendix B

Yale SCHOOL OF THE ENVIRONMENT

195 Prospect Street, New Haven, CT 06511
(203) 432-5100

January 26, 2022

Dr. Gary Carvalho
Proceedings of the Royal Society B

Dear Professor Carvalho, editorial team, and reviewers,

We sincerely appreciate the additional comments and suggestions to improve our manuscript, “Downtown Diet: a global meta-analysis of urbanization on consumption patterns of vertebrate predators” (RSPB-2021-2487). We have revised the text in order to address the points brought to our attention by the thoughtful reviewers.

Briefly, we have added a consideration of urban habitats as mosaics of both managed and natural habitat [Lines 81-83], increased prey abundance [Line 84], and pollution in urban habitats [Line 66] as suggested by reviewer 1. Summary information of HFI for the “wildland” category was added as well as a brief description on how the distinction between “rural” and “wildland” was made, following suggestions from reviewer 3. We have also adjusted grammar throughout the manuscript, such as revising the use of “due to” to “because of” when describing processes. Additional details on some key changes made to the manuscript as well as a brief response to each comment may be found below. We hope that these changes are satisfactory and improve our meta-analysis.

Reviewer 1: Concerning was how few studies were found in the southern hemisphere and that none were found for Africa. There definitely have been some urban diet studies of predators in African cities during this time. However, it is not clear if they only used studies where there was a comparison across a gradient. Also, many of the northern hemisphere studies focus on relatively few predators but have many studies on those species, and that is not reflected in your results.

Gámez et al: Thank you for this important comment. There was 1 study from Africa included in our analysis (Algeria), however we acknowledge the underrepresentation of studies from the African content. Although we did find studies on the diets of urban African predators, none satisfied the criteria for inclusion in our study (comparative urban *versus* non-urban framework). We further acknowledge the disparity in the representation of North American studies compared to African and Asian studies [Lines 404-407].

Reviewer 1: Did you use studies that presented these data only, or did you convert some of the other diet studies based on the typical isotopic signature? If the former, you need to present how many studies there were in your results.

Gómez et al: The former, we included studies which presented the isotope data in either graph or a raw data table format. We have added the number of isotope studies to the results [Line 278-279].

Reviewer 1: L254 over what period were 358 studies found? How many northern versus southern hemisphere studies? Why were only 32 then 44 included in your analyses?

Gómez et al: We have added detail on the year of publication for included studies [Lines 275-276] and additional consideration of the geographic bias to the discussion section. There were 32 studies in the initial search – we have revised the phrasing of this section of the text to improve clarity [Line 270].

Reviewer 1: I cannot access the Supplementary Table S1 but does that list all the species and the respective studies? Did you then group studies according to the respective vertebrate taxa? If yes, was that dominated by certain mammals or birds?

Gómez et al: The supplementary material lists studies grouped by taxa, not individual species, however we have added this information to the supplement. For mammals, there are 5 studies: 2 of which are on coyotes, 2 are on red foxes, and 1 on brown rats. For birds, there are 2 studies: 1 on white ibis, the other on the blue tit.

Reviewer 2: The most important is the title. “Diet” sounds too general. In my opinion the Authors should mention predation or predators, as it is the only trophic interaction studied in the work and also as the introduction begins with “predation.” This implies the importance of it and the further flow of the manuscript.

Gómez et al: Thank you for this comment, as it brings up a fruitful discussion the authors had on this point. An earlier version of the manuscript used the term “predation,” however, as we discussed the possibilities of how other resource acquisition pathways such as scavenging are at play in an urban environment, we broadened the language to “diet” to acknowledge that consumption does not equal predation.

Reviewer 2: I would like to see the captions for the supplementary material, there should table 1 and figures S1, S2, there is one file with graphs and one excel csv table

Gómez et al: Supplementary Figure 1: PRISMA diagram of literature search results and final inclusion of articles in the meta-analysis.

Supplementary Figure 2: Effect size (Hedges' g) for each study compared to corresponding HFI value of "wildland" and "rural" control site types for richness (top left), evenness (top left), Carbon IR (bottom left), and Nitrogen (bottom right).

Supplementary Table 1: Summary of each study included in the meta-analysis including sample size, predator taxa, variance, standard error, and effect size (Hedges' g).

Reviewer 3: The revisions have taken an already important and impactful paper and have made it even clearer. I appreciate the thoughtful responses of the authors to the most recent set of reviews and editorial comments. I think their additions to the methods and analyses fully address the reviewers' and editor's concerns.

Gómez et al: We continue to be appreciative of your thoughtful comments, constructive feedback, and support of the changes we have made to the manuscript.

Reviewer 3: Given these new inclusions, I think it would be helpful to make two additional clarifications (just a sentence needed for each):

- 1. Clarify what information was used for this additional categorization of sites as rural vs. wildland. Verbal description of the sites in the original study? examination of their maps?**
- 2. Include a summary of the HFI scores for the new rural vs. wildland categories.**

Gómez et al: Thank you for these suggestions we have added clarification to your first point [Lines 174-176] and have added HFI summary information for new control categories [Lines 315-320].

Reviewer 3: References on complexity in the category "rural" in case this is helpful:

Boone, C. G., C. L. Redman, H. Blanco, D. Haase, J. Koch, S. Lwasa, H. Nagendra, S. Pauleit, S. T. A. Pickett, K. C. Seto, and M. Yokohari. 2014. Reconceptualizing land for sustainable urbanity.

Padilla, B. J., and C. Sutherland. 2019. A framework for transparent quantification of urban landscape gradients. *Landscape Ecology* 34:1219–1229.

Gómez et al: These citations have now been added to the manuscript.